# Sun Declination and Distribution of Natural Beam Irradiance on Earth

José A. Rueda [1,*] , Sergio Ramírez [2,*], Miguel A. Sánchez [1] and Juan de Dios Guerrero [3]

1   Instituto de Agro-Ingeniería, Universidad del Papaloapan, Campus Loma Bonita, Oaxaca 68400, Mexico; msanchez@unpa.edu.mx
2   Facultad de Zootecnia y Ecología, Universidad Autónoma de Chihuahua, Chihuahua 31000, Mexico
3   Colegio de Postgraduados, Campus Puebla, Puebla 72760, Mexico; rjuan@colpos.mx
*   Correspondence: josearueda@yahoo.com (J.A.R.); srordonez@uach.mx (S.R.)

**Abstract:** The daily path of the Sun across longitude yields night and day, but the Sun also travels across latitude on a belt 47° wide. The solar meridian declination explains the latitudinal budget of *natural beam irradiance* (*NBI*), which is defined as the irradiance delivered to the Earth's surface as a normal projection from the Sun. Data for the Sun meridian declination were obtained from the Spencer model, known as the geometric model. The distribution of *NBI* was weighed for the latitudinal belt between the Tropics of Cancer and Capricorn. The variation in the parameters of solar meridian declination were found to be analogous to that of pendular motion. The joint distributions of the solar meridian declination against its own velocity, or that of the velocity against the acceleration of solar meridian declination, displayed circular functions. The *NBI* budget that a particular latitude gathers, fluctuates in inverse proportion to the velocity of solar meridian declination, yielding 18 sun-paths per degree for latitudes above 20°, or 6 sun-paths per degree of latitude for latitudes under 20°. At an average Sun–Earth distance of 1 AU, all sites of the planet, whose latitude coincides, whether within or between hemispheres, accumulate an equivalent budget of *NBI*.

**Keywords:** solar-heat distribution; solar-light distribution; Sun–Earth physics; normal irradiance; climate; *lumbra*; exposure term to *NBI*; resting term from *NBI*

## 1. Introduction

Solar radiation works as the unifying force that shapes all physical and biological elements of the planet [1,2]. The climate system is initiated by solar heat [3], while solar light fuels photosynthesis [2,3]. The annual cycle of solar declination is the primary factor driving the distribution of the Earth's climatic zones [3]. Small variations in solar irradiance scale climate responses globally [4]. For instance, surface temperatures rise 0.1 °C when the solar irradiance increases 0.1% [5]. The temperature of a site increases as the solar declination approaches the site's latitude, and decreases as the solar declination departs from that latitude [6].

In the sciences of energy, the global irradiance has usually been split into direct normal irradiance (DNI) and diffuse irradiance. DNI is defined as the radiation coming from the solar disk, received by a plate placed normal to the Sun [7], which is measured within a solid angle of up to 20° centered in the solar disk [7]. Because DNI can be assessed whenever a plate points to the Sun and wherever the Sun is visible, such a concept proves ineffective in explaining the latitudinal distribution of the Earth's budget of solar radiation; hence, the present work introduces several definitions aimed to analyze the irradiance supplied to the planet at the local meridian. *Natural beam irradiance* (*NBI*) denotes the share of the global irradiance that occurs as a normal projection to the Earth's surface when the solar disk (only 32 arcmin wide) occupies the local zenith. Unlike DNI, *NBI* is exclusive to the subsolar point and can only occur between the Tropics of Cancer and Capricorn, while it does not consider solid angles beyond the angular size of the Sun. *Natural oblique irradiance*

(*NOI*) denotes the share of the global irradiance that is supplied to a given site when the solar disk crosses the local meridian, but the solar declination lands elsewhere, so that *NOI* always lands at angles below 90°, while it can only occur at solar noon. *NOI* conforms to a line that pairs with the local meridian (where the center of such line holds *NBI*) and spans the entire Earth across every longitude within a solar day (≈24 h), following the planet's rotation. The angle at which *NOI* lands depends on the angular distance between the latitude holding *NBI* and the latitude where *NOI* is recorded. The obliquity angle of *NOI* remains unchanged for a given latitude throughout a solar day.

The angle at which the sunrays strike a site at the local meridian (*NOI* obliquity) fluctuates on a daily basis as a consequence of the annual cycle of solar declination. Every day, the Sun declination switches latitude, traveling the planet to a variable velocity. Conversely, the sunrays' absolute perpendicularity and its inherent *NBI* occur at a particular latitude only twice within a Gregorian year: when the solar declination converges that latitude on its way south, and then northward. Nonetheless, it can involve several instances within a season when the velocity of the solar meridian declination promotes the overlapping of the sun-paths. However, even on the days of the sunrays' perpendicularity, *NBI* occurs at a given longitude for merely 2.2 min, which is the time it takes for the apparent Sun to cross the local meridian from east to west: 15 arcdeg hour$^{-1}$, or 2.2 min for 32 arcmin (angular size of the Sun).

If the Sun, whose diameter is 109 times larger than that of the Earth, can cast an *umbra of light* on the subsolar point, the diameter of such *light umbra* might be comparable to the lunar umbra that appears during an annular eclipse. The given comparison is valid because of the equivalence in the angular sizes of the Sun and the Moon [8] as perceived from Earth. Despite the Sun's diameter being 400 times larger than that of the Moon, it is located 388 times farther from Earth than the satellite. Moreover, when the equatorial circumference of the planet is divided by 360°, it yields 111.3 km arcdeg$^{-1}$, which derives in a diameter of 60 km for the *lumbra*, given that the solar disk only covers 32 arcmin. The *great* disparity in diameters between the solar disk (1.392 × 10$^6$ km) and the *lumbra* (60 km) yields a ratio of 23,000:1. Accordingly, the bundle of sunrays that is normally projected from the Sun to the Earth takes the shape of a *cone of light*, which extends from the solar disk to the *lumbra*, and whose axis-vector extends from the center of the solar disk to the subsolar point.

That the *NBI* is significantly higher at the subsolar point is supported by several facts. (1) If light travels on a straight path [9], and a straight line is the shortest distance between two points (known facts), then the subsolar point is the closest spot to the Sun on the entire Earth [10]. (2) The sunrays that are delivered perpendicularly to the Earth cross the atmosphere through its shortest dimension [11], whereas the sunrays obliquely delivered interact with the atmosphere for tens of kilometers [12]. Finally (3), the Sun and the *lumbra* differ widely in diameters.

The Sun meridian declination defines the only latitude of the planet that receives the *light cone*, the *lumbra*, and therefore the *NBI*, within a period of 24 h, despite the Sun illuminating and heating half the globe at a time. When the apparent Sun reaches the zenith in the visible sky, the sunrays strike the land and the ocean at a right angle (90°), accounting for the highest power density received by a site throughout the year. As the power density of the sunrays decreases on par with their obliquity [13], the subsolar point [10] holds the highest *NBI* budget compared to any other site on the entire planet.

The Sun's radius is 696,340 km; hence, the sunrays coming from its edge are emitted at an increased distance of 696,340 km compared to those coming from the Sun's center (1.8 times the Earth–Moon distance). The sunrays from the center supply a higher concentration of heat and light to the Earth than those from the edge. The brightness of the solar disk declines from the core to the limb, while its temperature drastically drops beyond the chromosphere [14]. Every solar path, whose angle of declination approaches a given latitude, comprises a significant share of the total irradiance that a site accumulates throughout the year [7].

Because the sunrays can only reach normality at noon, the *NBI* must be assessed by means of the solar meridional declination. Assuming that the radiation intended for the Earth remains constant between two consecutive days (at a Sun–Earth distance of 1 UA), the budget of *NBI* can be assessed from the velocity of the Sun declination for the belt framed between the tropical parallels. The present work aimed to assess the budget of *NBI* for latitudes between the Tropics of Cancer and Capricorn, which has a close association with the budget of solar resources (heat and light) that every latitude can harness throughout the year. The working hypotheses are: (1) that the velocity and acceleration of the Sun meridian declination vary on par with the latitude of a site; (2) the velocity of solar meridian declination allows for an easy assessment of the annual budget of *NBI* for every latitude where *NBI* occurs; and (3) a known budget of *NBI* characterizes every particular range of latitude.

## 2. Materials and Methods

### 2.1. Data

Data for the Sun meridian declination ($\delta$) were generated from the geometric model of Spencer [15] (Equation (2)), which is valid for any Gregorian year. The Spencer model takes ground on the fractional year ($x$), in radians (Equation (1)), where the time $t$ is given as the day number within the year, from 1 to 365.

The velocity ($\omega$, arcmin day$^{-1}$) and acceleration ($\alpha$, arcsec day$^{-2}$) of solar meridian declination ($\delta$, arcdeg) are the first- and second-order derivatives (Equations (3) and (4)) of the solar meridian declination ($\delta$). Given that the Spencer model yields $\delta$ in radians, we switched the units of $\delta$, $\omega$, and $\alpha$ to arcdeg, arcmin day$^{-1}$, and arcsec day$^{-2}$, respectively, by adding the factors $[180/\pi]$, $\left[\frac{180}{\pi}\right]$ (60), and $[180/\pi]$ (3600) to Equations (2)–(4), in the same order. The three factors were used only after assessing the derivatives for $\omega$ and $\alpha$. To assess the derivatives, we followed the chain rule; for instance, the derivative $\partial\delta/\partial t$ was estimated by the product $(\partial\delta/\partial x)(\partial x/\partial t)$. The Equation of Time (**E**) was assessed for a one-year period, following the geometric model of Spencer [15] (Equation (5)). As **E** is also produced in radians, we included the factor $[180/\pi]$ [4] in Equation (5), in order to convert first from rad to arcdeg, and then from arcdeg to minutes (of time).

$$x = \left(\frac{2\pi}{365}\right)(t-1) \tag{1}$$

$$\delta = [0.006918 - 0.399912\,cos(x) + 0.070257\,sin(x) - 0.006758\,cos(2x) + 0.000907\,sin(2x) \\ -0.002697\,cos(3x) + 0.00148\,sin(3x)]\,[180/\pi] \tag{2}$$

$$\omega = (d\delta/dt) = \left[\tfrac{2\pi}{365}\right][0.399912\,sin(x) + 0.070257\,cos(x) + 0.013516\,sin(2x) + 0.001814\,cos(2x) \\ +0.008091\,sin(3x) + 0.00444\,cos(3x)]\,[180/\pi]\,[60] \tag{3}$$

$$\alpha = (d\omega/dt) = \left[\tfrac{2\pi}{365}\right]^2[0.399912\,cos(x) - 0.070257\,sin(x) + 0.027032\,cos(2x) - 0.003628\,sin(2x) \\ +0.024273\,cos(3x) - 0.01332\,sin(3x)]\,[180/\pi][3600] \tag{4}$$

$$\mathbf{E} = [0.0000075 + 0.001868\,cos(x) - 0.032077\,sin(x) - 0.014615\,cos(2x) - 0.040849\,sin(2x)]\,[180/\pi][4] \tag{5}$$

### 2.2. Variables

The Sun meridian declination ($\delta$) was assessed for every day of two non-leap Gregorian years to best represent the cyclical nature of the Sun-declination dynamics. The angular velocity ($\omega = \partial\delta/\partial t$) and acceleration ($\alpha = \partial\omega/\partial t$) of the solar meridian declination ($\delta$) were estimated for the same two-year period. Assessing the angular velocity and acceleration requires defining in advance adequate units for distance and time. We used arcdeg and day as the most natural units for distance and time, respectively. The given decision answers the fact that the geometric model yields a unique record of solar meridian declination for every single day of a Gregorian year.

*2.3. Defining Suitable Units*

Before applying the factors 60 and 3600, both already included in Equations (3) and (4), the records of $\delta$, $\omega$, and $\alpha$ took place in the range of $-23.5$ to 23.5 arcdeg, $-0.3896$ to 0.3953 arcdeg day$^{-1}$, and $-0.0069$ to 0.0078 arcdeg day$^{-2}$, respectively. After applying the given factors, the records of the three parameters $\delta$, $\omega$, and $\alpha$ fell in the range of 0 to 29; although the units were conveniently switched to arcdeg, arcmin day$^{-1}$, and arcsec day$^{-2}$, respectively.

Switching units served several purposes: (1) to avoid too small records lacking an integer component, (2) to represent the three parameters on a unified ordinate axis despite their differing dimensions, and (3) to illustrate the resemblance and associations between the functions $\delta$, $\omega$, and $\alpha$ when plotted first against time (days within a year), and then against the **E**.

*2.4. Declination Cycle*

The meridian declination ($\delta$), angular velocity ($\omega$), and acceleration ($\alpha$) of the apparent Sun were plotted against the Equation of Time (**E**); where **E** is defined as the difference between the mean time and the solar time and comprises the abscise of the Sun meridional analemma.

The signs were kept positive for every $\delta$ landing on the Northern Hemisphere, above the Equator (boreal spring and summer) as well as when $\omega$ corresponded to shifts of declination occurring between the Tropics of Capricorn and Cancer (boreal winter and spring), or when $\alpha$ corresponded to records of $\delta$ landing in the Southern Hemisphere (boreal autumn and winter), or when the **E** occurred to the right of the local meridian.

The signs were kept negative for every $\delta$ landing on the Southern Hemisphere, below the Equator (boreal autumn and winter) as well as when $\omega$ corresponded to shifts of declination occurring between the Tropics of Cancer and Capricorn (boreal summer and autumn), or when $\alpha$ corresponded to records of $\delta$ landing in the Northern Hemisphere (boreal spring and summer), or when the **E** occurred to the left of the local meridian.

Before combining the data of the two hemispheres, let us define the *net drive* (or *resultant drive)* within the dynamics of solar meridian declination. An accelerative *net drive* (speeding up) occurs when the product ($\omega\alpha$) yields a positive sign, whereas a decelerative *net drive* (slowing down) occurs when the product ($\omega\alpha$) yields a negative sign. As a rule, a season becomes accelerative when the solar declination approaches the Equator, whereas it becomes decelerative when the solar declination approaches either Tropical Parallel.

All signs were disregarded and data from both hemispheres combined because the signs indicated the direction of the *resultant drive* rather than magnitude. The meridian declination, velocity, and acceleration of the Sun were plotted against the **E**, obtaining three analemma-like shapes in the very same chart.

*2.5. Arbitrary Belts*

Five latitudinal belts were arbitrarily proposed on each hemisphere, from the Equator to either the Tropic of Cancer or Capricorn. Four belts (denoted *Equatorial*, A, B, C) framed 5 arcdeg each, while the fifth (denoted *Tropical*) framed only 3.5 arcdeg. The last belt was intentionally thinner to emphasize the fact that the *NBI* is expected to last longer on this belt.

Data were assessed within hemisphere. Later, all belts with equivalent limits of latitude were averaged together, disregarding the current direction of the cycle of solar declination or the hemisphere to which they belonged, because every belt had a corresponding belt in the opposite hemisphere regarding the latitudinal range. For instance, two sections of the declination cycle spanned the arbitrary belt denoted A (5, 10) during spring (5, 10] and summer [10, 5), while two sections spanned the corresponding range of latitude on the opposing hemisphere during autumn $(-5, -10]$ and winter $[-10, -5)$.

*2.6. Exposure and Resting Terms, and Budget of NBI*

As the Sun meridian declination cycle spans the very same arbitrary belt twice a year, from north to south and then backward, two additional concepts are proposed. The *exposure term* denotes the period on which an arbitrary belt holds *NBI*, whereas the *resting term* denotes the period on which an arbitrary belt lacks *NBI* because the solar declination cycle occurs elsewhere. The daily *budget of NBI* ($\Gamma_d$, Equation (6)) was assessed as the percentage of the Earth's annual budget.

$$\Gamma_d, \% \text{ day}^{-1} = [(1/365) * 100]\% = 0.274 \tag{6}$$

Equation (6) implies that every day within a Gregorian year, the planet receives 0.274% of its annual budget of *NBI*, assuming that the daily budget remains constant between successive days at an average Sun–Earth distance of 1 AU. By fixing the Sun–Earth distance at 1 AU, we intentionally dismissed its role on the *NBI*, which serves the purpose of isolating the effect of solar meridian declination as a key explanatory variable for the distribution of *NBI*. The result of Equation (6) accounts for the relative value of a single sun-path among the 365 sun-paths occurring throughout the year. The relative budget of *NBI* harnessed by an arbitrary belt (% arcdeg$^{-1}$) was assessed following Equation (7), where the exposure term is the number of days that the Sun lasts on every given belt.

$$\Gamma_b, \% \text{ arcdeg}^{-1} = \left[ (\text{Exposure term of the belt, day}) \left( \Gamma_d, \% \text{ day}^{-1} \right) \right] / (\text{Belt width, arcdeg}) \tag{7}$$

As $\omega$ is the vertical distance between two consecutive records of solar declination, $\omega$ can be thought as the width of the belt on which the daily budget of *NBI* ($\Gamma_d$) is distributed within a day. The accumulated budget of heat and light delivered to a particular arbitrary belt, as *NBI*, varies in association with the $\omega$. The integrated budget of *NBI* available for every latitude (% arcdeg$^{-1}$) was assessed for belts of size $\omega$ and an *exposure term* of a single day, which yielded Equation (8). Equation (8) incorporates the factor 60 in order to express the results as a budget of *NBI* per arcdeg of latitude instead of arcmin.

$$\Gamma, \% \text{ arcdeg}^{-1} = 60 \, \Gamma_d / \omega \tag{8}$$

**3. Results**

*3.1. Declination Cycle*

The functions of meridian declination ($\delta$, arcdeg), angular velocity ($\omega = \partial\delta/\partial t$, arcmin day$^{-1}$), and acceleration ($\alpha = \partial\omega/\partial t$, arcsec day$^{-2}$) of the Sun against time (Figure 1) vary within a similar range, as if they represent the very same variable (except for the wider amplitude of $\alpha$); therefore, a record of $\omega$ that matches that of the current $\delta$ occurs about three months earlier, whereas a record of $\alpha$, that matches that of the current $\omega$, occurs about three months before; both given distances signify the length of an entire season. The functions of $\delta$, $\omega$, and $\alpha$ yield pseudo-sinusoidal curves that resemble each other in shape, amplitude, and frequency (Figure 1).

We applied the factors 60 and 3600 to the definition of the parameters given in Equations (3) and (4), in order to show the remarkable resemblance between the three functions ($\delta$, $\omega$, and $\alpha$) when plotted against time; nonetheless, we must keep in mind that the actual size of the $\omega$ function falls within a range of 1/60th of the interval spanned by $\delta$, whereas that of $\alpha$ falls within a range of 1/60th of the interval spanned by $\omega$. Therefore, $\alpha$ takes place within a range of 1/3600th of the range covered by $\delta$.

Following the parameters defined in Equations (2)–(4), the functions $\delta(\omega)$ and $\alpha(\omega)$ approximate circumference-like shapes. The association between $\omega$ and $\delta$ approaches the circle displayed in Figure 2 (Equation (9)), whereas the association between $\omega$ and $\alpha$ approaches the circle displayed in Figure 3 (Equation (10)). Nevertheless, the right side of Equations (9) and (10) does not conform to meaningful units for the radii $r_1$ and $r_2$. Therefore, the circles $\delta(\omega)$ and $\alpha(\omega)$ arise only as numerical associations between $\omega$ and $\delta$, or between $\omega$ and $\alpha$, in accordance with the definitions in Equations (2)–(4).

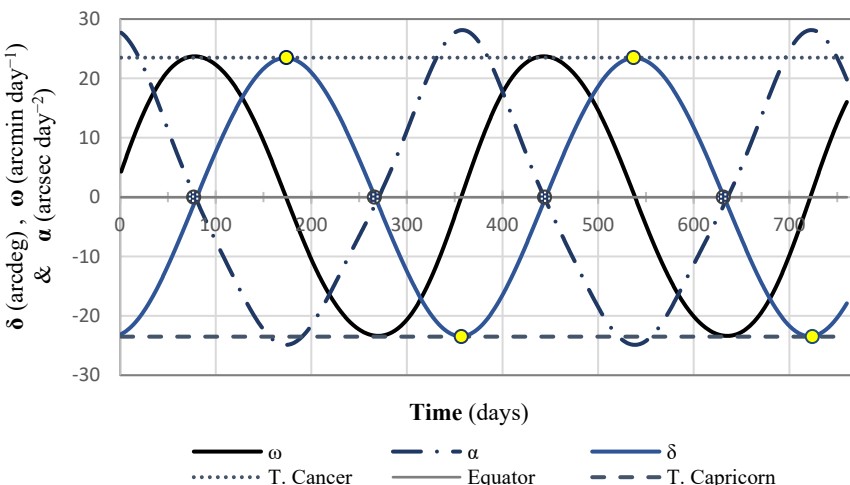

**Figure 1.** Meridian declination ($\delta$), angular velocity ($\omega = \partial\alpha/\partial t$), and angular acceleration ($\alpha = \partial\omega/\partial t$) of the Sun for two subsequent Gregorian years built from the geometric model of solar declination. Equinoxes are tagged on the Equator (solid circle) and solstices on the Tropics of Cancer and Capricorn (open circle).

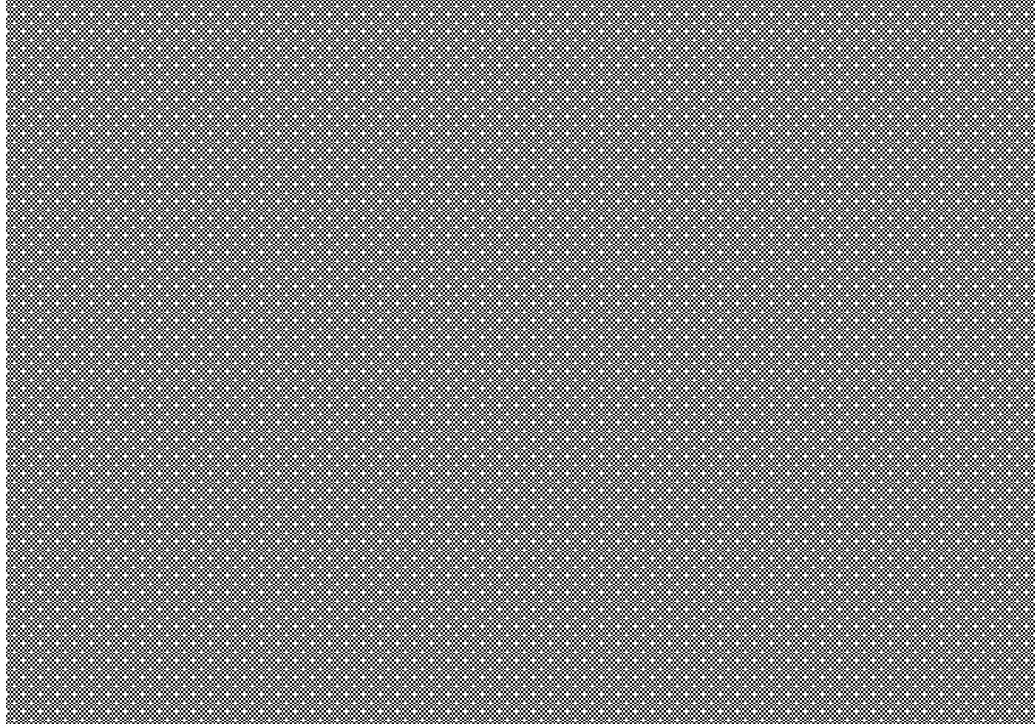

**Figure 2.** Sun meridian declination ($\delta$) against the velocity of declination ($\omega = \partial\delta/\partial t$) throughout a Gregorian year. Equinoxes (solid circle) are tagged in the Equator and solstices (open circle) in the Tropics of Cancer and Capricorn.

According to Figures 2 and 3, which agree with Equations (9) and (10), respectively, the highest $\alpha$ can only occur at the lowest $\omega$ and vice versa, whereas the same holds true for the association between $\delta$ and $\omega$. Given that the sinusoidal amplitude of $\alpha$ diverges from the amplitudes of $\delta$ and $\omega$ (Figure 1), the chart in Figure 3 deviates from a flawless circumference. For instance, while the radius $r_1$ in Equation (9) (Figure 2) fluctuated from 23.2 to 24.3 and averaged 23.6, the radius $r_2$ in Equation (10) (Figure 3) fluctuated from 23.0 to 28.1 and averaged 24.5.

$$\omega^2 + \delta^2 = r_1{}^2 \tag{9}$$

$$\omega^2 + \alpha^2 = r_2{}^2 \tag{10}$$

where $\delta$ is the solar meridian declination (arcdeg); $\omega$ is the velocity of solar meridian declination (arcmin day$^{-1}$); $\alpha$ is the acceleration of solar meridian declination (arcsec day$^{-2}$); $r_1$ and $r_2$ are the radii.

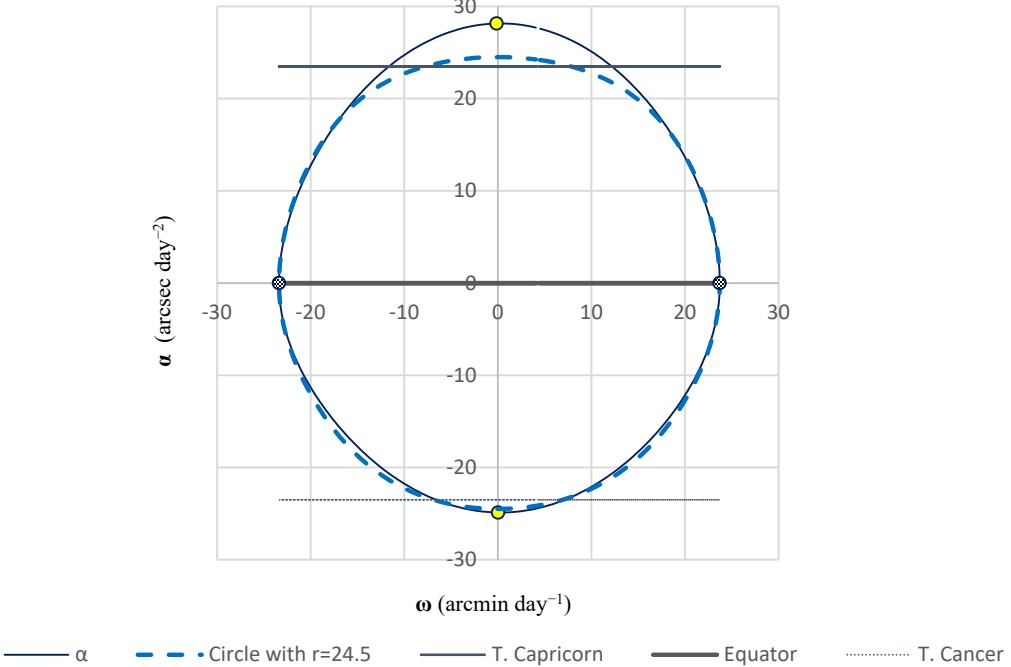

**Figure 3.** Acceleration of declination ($\alpha = \partial\omega/\partial t$) against the velocity of solar meridian declination ($\omega = \partial\delta/\partial t$) throughout a Gregorian year. Equinoxes (solid circle) are tagged in the Equator and solstices (open circle) in the Tropics of Cancer and Capricorn.

The correlation coefficients occurring between $\omega$ and $\delta$ exhibited an alternated direct/inverse association (r = $\pm$0.91; $p < 0.001$), where the sign of the coefficient remained unchanged within a season but switched between seasons. Analogously, the correlation coefficients occurring between $\omega$ and $\alpha$ exhibited an alternated direct/inverse association (r = $\pm$0.93; $p < 0.001$), where the sign of the coefficient remained unchanged within a season but switched between seasons. An analysis based entirely on the linear associations would have built rhomboidal charts instead of the circumference-like functions shown in Figures 2 and 3, corresponding to Equations (9) and (10). Following the linear correlations as well as the data in Figures 2 and 3, the association between $\delta$, $\omega$, and $\alpha$ is clear: $|\omega| \propto 1/|\delta|$, $|\omega| \propto 1/|\alpha|$, so that $|\alpha| \propto |\delta|$. The first two associations were displayed in Figures 2 and 3, whereas the third relation approached a straight line; the numerical values of the three parameters ranged from 0 to 29.

### 3.2. Analemmatic Charts

The functions of $\delta$, $\omega$, and $\alpha$ over the **E** are shown in Figure 4. Most $\alpha$ records displayed in quadrants I and II resembled, mirrored, and corresponded to the values of $\delta$ shown in quadrants III and IV, respectively. Furthermore, most $\omega$ records displayed in quadrant II and IV corresponded to spring and summer, while most $\omega$ records shown in quadrants III and I corresponded to autumn and winter.

When $\delta$, $\omega$, and $\alpha$ were plotted against the **E** (Figure 4) the three analemmatic curves depicted lemniscate-shapes whose records fell within a similar range. Since $\omega$ and $\alpha$ were scaled up through the incorporation of the factors 60 and 3600, the actual sizes of the $\omega$ and $\alpha$ lemniscates were of 1/60th and 1/3600th, respectively, compared to the analemma ($\delta$). Unlike $\delta$ and $\alpha$, the sections of the $\omega$ lemniscate falling into quadrants I

and II enclosed a similar area to that area enclosed by quadrants III and IV of the same lemniscate. Furthermore, the $\omega$ lemniscate slanted to the left.

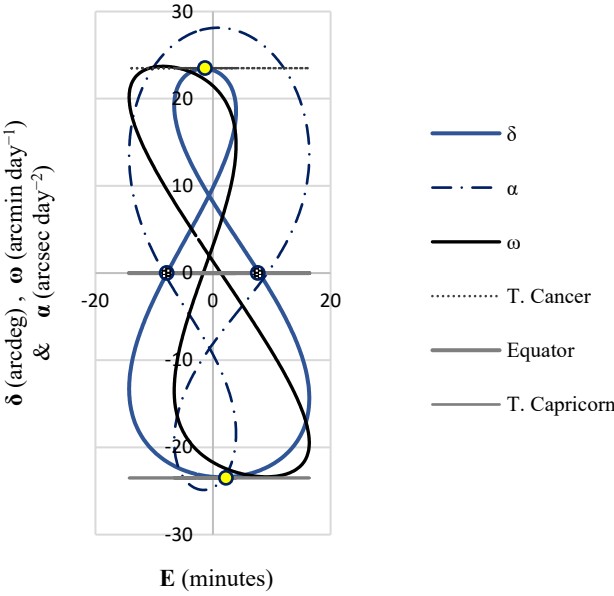

**E** (minutes)

**Figure 4.** Angular velocity ($\omega = \partial\delta/\partial t$) and angular acceleration ($\alpha = \partial\omega/\partial t$) of the Sun meridian declination ($\delta$) in association with the Equation of Time (**E**, minutes of time) within a Gregorian year. Equinoxes (solid circle) and solstices (open circle) are tagged on the analemma ($\delta$) at the three main parallels.

### 3.3. Budget of NBI

The solar meridian declination, *exposure term*, *resting term*, and the budget of *NBI* ($\Gamma_b$, % arcdeg$^{-1}$) are shown in Table 1 for each arbitrary belt. In Table 1, the two sections of the Sun declination cycle that provide the *NBI* to each arbitrary belt are presented separately. The same information presented in Table 1 is shown in Table 2, where the two *exposure terms* of each particular belt are combined and the data were averaged for both hemispheres.

A lower $\omega$ yields a higher *exposure* to *NBI*, therefore the budget of solar heat and light that a site holds increases on par with its latitude from either *Equatorial* belt to the corresponding *Tropical* belt (Table 1). The signs of $\delta$, $\omega$, and $\alpha$ are always consistent within season, but they do switch between seasons. Hence, the same *resultant drive* characterizes an entire season, while the next season always presents a contrasting *resultant drive*. Seasons whose *resultant drive* is accelerative are characterized by holding equal signs for $\omega$ and $\alpha$, whereas seasons whose *resultant drive* is decelerative are characterized by holding opposing signs for $\omega$ and $\alpha$.

In every season, the path of the Sun goes through the five arbitrary belts of a hemisphere, whereas the average records of $\delta$, $\omega$, $\alpha$, and $\Gamma_b$ exhibit a close symmetry both between seasons and between hemispheres. In the Northern Hemisphere, the arbitrary belts denoted as *Equatorial*, A, B, C and *Tropical* hold *exposure terms* of 24, 29, 30, 37, and 66 sun-paths (days), respectively, throughout the year. These values represent 6.57, 7.94, 8.22, 10.14, and 18.08% of the Earth's annual budget of *NBI*, equivalent to 1.32, 1.59, 1.64, 2.03, and 5.17% arcdeg$^{-1}$, respectively.

In the Southern Hemisphere, the arbitrary belts denoted as *Equatorial*, A, B, C, and *Tropical* hold *exposure terms* of 26, 26, 30, 37, and 60 sun-paths (days), respectively, throughout the year. These values represent 7.12, 7.12, 8.22, 10.13, or 16.43% of the Earth's annual budget of *NBI*, equivalent to 1.42, 1.42, 1.64, 2.03, and 4.70% sun-paths arcdeg$^{-1}$, respectively.

**Table 1.** Angular velocity ($\omega = \partial\delta/\partial t$) and acceleration ($\alpha = \partial\omega/\partial t$) of the Sun meridian declination ($\delta$) as well as within season records for *exposure term* and the budget of *natural beam irradiance* attained by each of the ten *arbitrary belts* ($\Gamma_b$).

| Season | Belt Name | $\delta$ (arcdeg) | $\omega$ (arcmin day$^{-1}$) | $\alpha$ (arcsec day$^{-2}$) | Exposure Term (day) | Starting Date (month/day) | $\Gamma_b$ (% arcdeg$^{-1}$) |
|---|---|---|---|---|---|---|---|
| Spring | Equatorial | [+0, +5) | 23.48 | −3.00 | 11 | 3/22 | 0.60 |
| | A | [+5, +10) | 22.32 | −7.38 | 15 | 4/2 | 0.82 |
| | B | [+10, +15) | 19.86 | −12.30 | 15 | 4/17 | 0.82 |
| | C | [+15, +20) | 15.74 | −17.47 | 18 | 5/2 | 0.99 |
| | Cancer | [+20, +23.5] | 6.62 | −23.27 | 34 | 5/20 | 2.66 |
| Summer | Cancer | (+23.5, +20] | −6.59 | −23.11 | 32 | 6/23 | 2.50 |
| | C | (+20, +15] | −15.41 | −17.21 | 19 | 7/25 | 1.04 |
| | B | (+15, +10] | −19.57 | −11.93 | 15 | 8/13 | 0.82 |
| | A | (+10, +5] | −21.91 | −7.36 | 14 | 8/28 | 0.77 |
| | Equatorial | (+5, +0) | −23.09 | −3.09 | 13 | 9/11 | 0.71 |
| Autumn | Equatorial | [+0, −5) | −23.30 | 1.24 | 13 | 9/24 | 0.71 |
| | A | [−5, −10) | −22.52 | 6.00 | 13 | 10/7 | 0.71 |
| | B | [−10, −15) | −20.46 | 11.66 | 15 | 10/20 | 0.82 |
| | C | (−15, −20] | −16.11 | 18.68 | 19 | 11/4 | 1.04 |
| | Capricorn | [−20, −23.5] | −6.67 | 26.11 | 30 | 11/23 | 2.35 |
| Winter | Capricorn | (−23.5, −20] | 6.90 | 26.26 | 30 | 12/23 | 2.35 |
| | C | (−20, −15] | 16.78 | 18.57 | 18 | 1/22 | 0.99 |
| | B | (−15, −10] | 20.64 | 12.24 | 15 | 2/9 | 0.82 |
| | A | (−10, −05] | 22.82 | 6.36 | 13 | 2/24 | 0.71 |
| | Equatorial | (−5, +0) | 23.64 | 1.30 | 13 | 3/9 | 0.71 |

Declination data were derived from the geometric model of solar declination [15]. The five belts of spring are the very same five belts of summer. $\Gamma_b = \left[ (\text{Exposure term, day}) \left( \Gamma_d, \text{\% day}^{-1} \right) / (\text{Belt width, arcdeg}) \right]$.

**Table 2.** Angular velocity ($\omega = \partial\delta/\partial t$) and acceleration ($\alpha = \partial\omega/\partial t$) of the Sun meridian declination ($\delta$), *exposure*, and *resting terms*, as well as the budget of the *natural beam irradiance* of each arbitrary belt ($\Gamma_b$), averaged for two hemispheres throughout a Gregorian year.

| Arbitrary Belt | $\delta$ (arcdeg) | $\omega$ (arcmin day$^{-1}$) | $\alpha$ (arcsec day$^{-2}$) | Exposure Term (days) | Resting Term 1 (days) | Resting Term 2 (days) | $\Gamma_b$ (% arcdeg$^{-1}$) |
|---|---|---|---|---|---|---|---|
| Equatorial | (0, 5) | 23.4 | 2.2 | 25.0 | 157.5 | 182.5 | 1.37 |
| A | [5, 10) | 22.4 | 6.8 | 27.5 | 130 | 207.5 | 1.51 |
| B | [10, 15) | 20.1 | 12.0 | 30.0 | 100 | 235 | 1.64 |
| C | [15, 20) | 16.0 | 18.0 | 37.0 | 63 | 265 | 2.03 |
| Tropical | [20, 23.5] | 6.7 | 24.7 | 63.0 | 0 | 302 | 4.93 |
| | Weighted average | 15.4 | 15.4 | | | | |

$\Gamma_b$ involves the sum of *NBI* budgets for the two exposure terms on which the belt holds the *NBI* along the year.

The daily *budget of NBI* ($\Gamma$, % arcdeg$^{-1}$) delivered per arcdeg of latitude, assessed as a fraction from the annual budget intended for the planet (Equation (4)), is shown in Figure 5. The *NBI* budget of a belt can also be thought of as the quotient between its *exposure term* (in days) and the number of days in a year. The *NBI* budget intended for each *Tropical* belt (between 20 and 23.5°) is received during a unique *exposure term* of 66 or 60 days, which is in turn followed by a unique *resting term* 299 or 305 days long, for the Northern or Southern Hemisphere, in the same order. The belts A, B, and C hold two *exposure terms* of equal length as well as two *resting terms* of contrasting length, where the shorter the first *resting term* the longer the second.

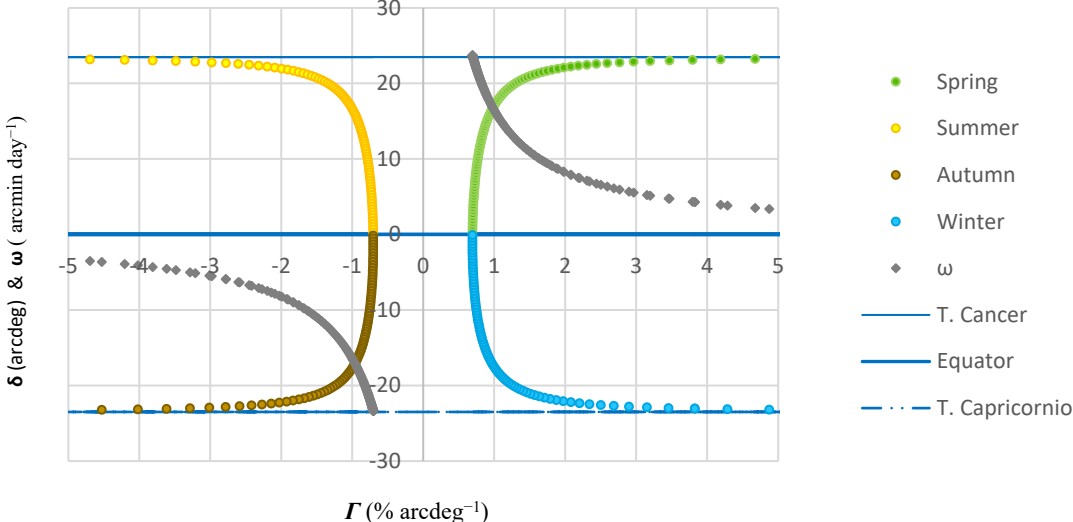

**Figure 5.** Budget of *natural beam irradiance* ($\Gamma$) expressed as a percentage of the budget annually available (100%) that is delivered per arcdeg of latitude, where $\Gamma = 60\ \Gamma_d / \omega$ and $\Gamma_d = [(1/365\ \text{day}) * 100]\% = 0.274\%\ \text{day}^{-1}$. The function $\Gamma$ is presented separately for each season (green, yellow, brown, and blue circles).

In Figure 5, $\Gamma$ ranged from 0.69 to 4.0 for 327 out of the 365 days, but grew higher during the 20 days centered in the summer solstice or 18 days centered in the winter solstice. Illogical records occurred near the solstices, which implies that $\omega$ is so slow that many sun-paths overlap on the very same latitude. The total $\Gamma_b$ that each arbitrary belt gathers along the year (Table 2) is the sum of the budgets of the two seasons on which the Sun declination cycle spans the very same range of latitude (Table 1). In Figure 5, the budgets of *NBI* for summer and autumn are displayed as "negative records", despite the fact that they cannot be negative, and they are not; however, the chart takes advantage of the sign of $\omega$ to distinguish between the seasonal budgets of *NBI* supplied in summer vs spring or between autumn vs winter.

## 4. Discussion

The structure of the information shown in Figures 1–5 depends on the units chosen for $\omega$ and $\alpha$. Likewise, approximating the diameter of the circular shapes $\delta(\omega)$ and $\alpha(\omega)$ of Equations (9) and (10) (Figures 2 and 3), respectively, can only be possible by scaling up the velocity and acceleration of the solar meridian declination by the factors 60 and 3600. The similarity in range between the three parameters of solar declination, once scaled up, implies that the actual velocity of solar meridian declination occurs at a rate of 1/60th, and the actual acceleration of solar meridian declination occurs at a rate of 1/3600th, both compared to the range in which $\delta$ falls.

The variation found in $\delta$, $\omega$, and $\alpha$ is typical of oscillatory pendular motion. The annual cycle of solar declination features two equilibrium points at the equinoxes and two resting points at the solstices. A trough of $\omega$ converges to a peak of $\alpha$ at the solstices, where the Sun slows down, comes to a standstill, and then speeds up during the days preceding, the day at, and the days following a solstice, respectively. A trough of $\alpha$ converges to a peak of $\omega$ at the Equator, where $\alpha$ progressively decreases, reaches zero, and then progressively increases during the days before, the day at, and the days after an equinox, respectively. The apparent Sun progressively brakes during spring or autumn as it approaches a solstice, while it shows diminishing $\alpha$ all through summer or winter.

The association between $\delta$ and $\omega$ fits a circumference, where spring, summer, autumn, and winter lie in the quadrants I, II, III, and IV of the circle $\delta(\omega)$, respectively. When signs are disregarded, $\delta$ varies in direct proportion to $\alpha$. Every range of latitude conforms to characteristic records of $\delta$, $\omega$, and $\alpha$. For example, the four instances of *belt C* showed com-

parable records of $\omega$ and $\alpha$, with only slight differences between hemispheres. Disregarding the signs, the dynamics of declination was equivalent between seasons whose *resultant drive* coincide (spring vs autumn; summer vs winter), whereas the three parameters $\delta$, $\omega$, and $\alpha$ varied in reverse order between seasons whose *resultant drive* differed (spring vs. summer; autumn vs. winter).

According to the Sun meridian declination, every arbitrary belt holds *NBI* during two *exposure terms* within a year; both of equal length and each followed by a characteristic *resting term*. The first *exposure term* occurs when the Sun meridian declination spans an arbitrary belt on its way north, and the second *term* occurs when the solar declination spans the very same belt on its way south. For the *Tropical belt*, the two *exposure terms* merge into a unique *exposure term*, and the two *resting terms* merge into a unique *resting term*. Conversely, the two *exposure terms* of any belt centered at the Equator would have the same length, while the same holds true for the two *resting terms*.

For the average of both hemispheres, the belts A, B, C, and *Tropical* accumulated 1.1, 1.2, 1.5, and 3.6 times the budget of the *Equatorial* belt throughout the year (arcdeg$^{-1}$). When $\Gamma_b$ was multiplied by the width of each arbitrary belt (arcdeg), the results yielded 6.9, 7.5, 8.2, 10.1, and 17.3% of the annual budget of *NBI* for the belts *Equatorial*, A, B, C, and *Tropical*, where the five percentages added up to 50%, and corresponded to the average *exposure terms* of 5.0, 5.5, 6.0, 7.4 and 18 sun-paths arcdeg$^{-1}$, respectively.

The low $\omega$ of the *Tropical* belt guarantees a high budget of *NBI*, however irregularly supplied. The long *exposure term* of the *Tropical* belt also secures a high budget of *NOI* for latitudes beyond the Tropics of Cancer and Capricorn, which is provided at the lowest obliquity possible.

The highest temperatures on Earth fall on latitudes in the vicinity of the Tropics of Cancer or Capricorn during their *exposure* to *NBI*. A high temperature has been recorded for latitudes around the Tropics of Cancer and Capricorn [16] during their high *exposure terms* to both *NBI* and *NOI* of low obliquity. In the Northern Hemisphere, ocean temperatures for latitudes $\geq 20°$ and depths from 0 to 50 m increase steadily throughout spring and summer, reaching their peak just before *NBI* leaves the hemisphere (equinox); conversely, for the equatorial latitude to reach their peak temperatures, a heating cycle is required including near-perpendicular *NOI* approaching *NBI*, followed by *NBI* itself, and finally near-perpendicular *NOI* departing from *NBI* [17]. A different study found that temperatures were lower in the Southern Hemisphere, both on sea and land compared to those of the Northern Hemisphere [18]. This disparity might be explained by the oceans' ability to store heat, given the larger share of oceans in the Southern Hemisphere.

The high budget of *NBI* of the *Tropical* belt might be the cause of the location of the latitudinal deserts of the globe because most deserts occur near the Tropics of Cancer and Capricorn. The location of altitudinal deserts may also be related to their budget of *NBI*, because both landscapes and hill slopes may occur naturally "tilted to the Sun". In high altitudes, the sunrays cross the atmosphere through a thinner, cooler, lighter, drier, and unpressured air-layer [19]; therefore, the net radiation reaching the land might be higher compared to sites of low altitude. One previous work proposed that the deserts were originated by disturbances in the water cycle, induced by both natural causes and anthropogenic activities [20].

The lowest *NBI* budget of the inter-tropics is seized by the Equator, however, this is compensated by a daily dosage of *NOI* of low obliquity (incidence angle above 66.5 arcdeg) along the year. Being at the center of the declination cycle, the Equator secures a constant budget of natural irradiance throughout the year. The number of solar paths delivering *NOI* to the Equator increases on par with the obliquity of their beam. The $\omega$ recorded in the *Equatorial* belt was 3.5 times higher than that of the *Tropical* belt, whereas the *Tropical* belt received a 3.6 higher budget of *NBI* than the *Equatorial* belt. Given the symmetry in the budget and distribution of *NBI* between hemispheres, every pair of sites matching latitudes, whether in the same or opposing hemispheres, has the potential to foster similar climates. Nonetheless, there were slight differences between hemispheres; for instance,

the northern *Tropical* belt held 18.85 sun-paths arcdeg$^{-1}$, whereas the southern only held 17.15 sun-paths.

The distribution and budget of *NBI* may be associated to the location of both the ITCZ and the rainy belt [21]. The contrasting temperatures occasioned by inter-seasonal variations in the obliquity of the sunrays might be associated with the occurrence of hurricanes, cyclones, and typhoons in both the *Tropical* belts and latitudes surrounding them (15 to 30°) [22].

Apart from solar declination, two factors promoted variations in the budget and distribution of *NBI* across latitude. The first was the higher use-efficiency of the solar resources given by the reduction in the parallels' length as the latitude increased. The spheric shape of the Earth yielded a lower linear velocity for any sun-path traveling across the longitude on a belt of higher latitude. The average within-day *term* in which the *Equatorial*, A, B, C, and *Tropical* belts held *NBI* was 0.5, 2.5, 5.5, 8.7 or 9% higher, respectively, compared to latitude zero. The second factor was the solar constant, whose variation was flawlessly synchronized with both the solar declination and the Earth's revolution. The solar constant averaged 1361 Wm$^{-2}$ [4], but applying the inverse law of light [23] to the Sun–Earth distances along the year 2024 [24], it ranged from 1316 to 1407 Wm$^{-2}$ (5 July and 3 January 2024, respectively), while its association with the solar declination followed Equation (11) ($R^2 = 0.95$, $p < 0.0001$, *se* of $\beta_1 = 0.02$). Hence, the Southern Hemisphere holds a budget of *NBI* that is 4.2% higher than that of the Northern.

$$Solar\ constant = 1361.9 - 1.9\ \delta \tag{11}$$

The planet's budget of *NBI* is unevenly distributed across latitude, which might bring new insights and applications in the fields of solar energy, but might also imply some long-term negative consequences for the environment. For instance, one-third of the planet's budget of *NBI* lands on two thin belts, 3.5 arcdeg wide (20–23.5°), one on each hemisphere. To start with, these *energy belts* are bands of latitude where the harvest of solar energy might be highly efficient. Conversely, despite such enormous concentration of *NBI* guaranteeing a high budget of low-obliquity *NOI* for latitudes beyond the Tropics of Cancer or Capricorn, the inherent high dose of solar heat might play a key role explaining the growing desertification of the globe. The latter effect is aggravated by the absence of tree cover that already characterizes a large fraction of these *energy belts* (20 to 23.5°).

## 5. Conclusions

The dynamics of the Sun meridian declination is analogous to that of pendular motion, with two equilibrium points at the equinoxes and two resting points at the solstices, respectively. The highest velocity converges to a trough of null acceleration at the equinoxes, whereas the highest acceleration converges to a trough of null velocity at the solstices. The annual cycle solar meridian declination behaves monotonically accelerative throughout summer and winter, and monotonically decelerative throughout spring and autumn.

The velocity of the Sun meridian declination modulates the distribution of the solar resources, heat and light, on the surface of the planet. The lower the velocity, the higher the budget of *NBI*. A high budget of solar resources is delivered to the latitudinal belt 20–23.5° within a two-month *exposure term*, which is followed by a ten-month *resting term*. Near equatorial latitudes hold the lowest budget of *NBI* within the inter-tropics and the highest budget of *NOI*. The budget of *NBI* delivered to every latitude throughout the year depends on the cycle of solar declination. According to the latitudinal budget of NBI, equivalent latitudes foster sister climatic zones, while divergent latitudes foster contrasting climatic zones, both within or between hemispheres.

**Author Contributions:** Conceptualization, J.A.R.; Methodology, J.A.R. and S.R.; Formal analysis, J.d.D.G., J.A.R. and M.A.S.; Investigation, J.A.R. and J.d.D.G.; Writing—original draft preparation, J.A.R., J.d.D.G. and M.A.S.; Writing—review and editing, S.R.; Visualization, J.A.R.; Supervision,

J.A.R. and S.R.; Project administration, J.A.R. All authors have read and agreed to the published version of the manuscript.

**Funding:** This research received no external funding.

**Institutional Review Board Statement:** Not applicable.

**Informed Consent Statement:** Not applicable.

**Data Availability Statement:** Data generated for this manuscript was uploaded to: http://doi.org/10.6084/m9.figshare.19970816 (last-time accessed on 10 August 2020).

**Conflicts of Interest:** The authors declare no conflicts of interest.

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
