# Peer review of "Sun Declination and Distribution of Natural Beam Irradiance on Earth"

_atmosphere, doi:10.3390/atmos15081003_

Round 1
Reviewer 1 Report
Comments and Suggestions for Authors
Please check the attached file.

| The authors make serious errors in equations, and in particular in units, which if this had been a high school exam, would lead to an F: Fail |
Author Response
R1: Interesting insights. Every comment was analyzed and changes were conducted accordingly. The results were double checked and some slight changes were required given the formalization of Math.
R2: A profound review. Deep insights. Also, interesting ideas to keep on in the same line of research.
R3: Gets the transcendence of the work, obliged to a brief connection to topics on Energy and the Environment. Done!

Reviewer 2 Report
Comments and Suggestions for Authors
In this version of "Sun declination and distribution of natural beam irradiance on Earth" by José A. Rueda et al., the authors have substantially revised and improved the manuscript.
In my view, this article only supports the basic understanding of the geometric relationship between the Sun and the Earth , and at this level, it cannot be published without major revisions.
Here are two main comments for your reference.
1. Existing software applications are already capable of performing similar calculations and analyses, such as ArcGIS, even including terrain-specific assessments. The question arises: what distinguishes the algorithm introduced in this paper from existing software?
2. The conclusion is weak, lacking significant scientific findings, and it fails to provide concrete support for any specific application.
The reviewer tried to propose some potential directions for this ms.
1. Validate the model by integrating ground-based observational data or satellite observational data to confirm the model's effectiveness.
2. Calculate the solar irradiance of other planets and conduct comparative analysis to demonstrate from the perspective of solar irradiance that Earth is a more habitable planet.
Author Response

(The authors gave the same response as above.)

Reviewer 3 Report
Comments and Suggestions for Authors
This paper intitled " Sun declination and distribution of natural beam irradiance on Earth" presents an analysis of the Sun's movement across both longitude and latitude.
Also, the paper provides important insights into the Sun's latitudinal movement and its impact on budget of natural beam irradiance (NBI) on Eath surface. By connecting meridian declination to NBI distribution, the study deepens our understanding of solar energy patterns.
Finally, I suggest that the author discuss the interesting findings in this paper (for exemple by adding a subsection in result section) by exploring their practical applications or implications for fields such as solar energy and climate studies.
Author Response

(The authors gave the same response as above.)

Round 2
Reviewer 1 Report
Comments and Suggestions for Authors
It becomes clear that there is a scaling problem regarding the units they use.
Since units have to be defined precisely in a scientific paper, I cannot recommend this paper for publication before this is solved.
Attached are my comments, with the most serious marked in red.

No comments
Author Response
Our unique reviewer of Round 2 posed some strong comments which oblige to double check some Math. By such comments R1 even tested its own line of thought already included in the manuscript in order to force to an in-depth last scrutiny. R1 challenges the authors to stand for and defend the ideas and modest models as included in the actual version. Our referee led us to verify the variables defined and the correctness of the whole variable-equation-unit agreement. Most of the analyses and challenges R1 posed did not land to any actual alteration on the manuscript; but the annotations were valuable and useful to double check most of methods and results.
We became somehow lost on some comments when no specific changes are suggested, but rather imply some observational commentaries. But asportations shall be sound and grounded. Conversely, the shifts which were punctually addressed were proven ineffective.

Round 3
Reviewer 1 Report
Comments and Suggestions for Authors
In the previous review I raised two objections as requirement for me to recommend publishing:
The first was as scale change of 360 (a circle) to 365 (the length of year in days). I find the authors have argued well for their choice of 360 and I will withdraw my objections.
Further I agree with the authors that the Spencer model is based on an elliptical orbit. This is seen in figure 1 where the length of the seasons is different. The season with positive declination is about 6 days longer than the season of negative declination. This is also shown in Spencer’s equations where the higher order terms show the deviation from a cirle.
The second requirement was to delete equations 9 and 10. This is not done.
For this reason, I cannot recommend the paper to be published.
The equations 9 and 10 as they are written are wrong. You cannot add elements with different units.
Eq9: (60nu)2 + ∂2 ≈ r2. (angle/time)2 + angle2 = r2, no meaningful unit for r
Eq10: 60 nu2 + (3600phi)2 =r2. (angle/time)2 + (angle/time2)2 = r2, no meaningful unit for r
(Sorry I only have a few greek letter in this Word).
I suggest a possible solution for you:
Your point is that by selecting proper units for the three variables: angle, velocity and acceleration of the declination (arcdeg, arcmin, and arcsecond) you get numbers which let you show the three variables in the same graph. If you stay with these units, it is not necessary to introduce new units nu and phi
I strongly recommend keeping the units omega, alpha and delta as you have in figure 1 and not introduce the 1/60 and 1/3600 times units
You want to show in eqs. 9 and 10 that with proper units, you can make a sum of the numerical values squared which is about the same (circle in fig 2). In mathematics there is a sign for numerical values which consists of bars on both side of the element :
l alpha l – (sort I don’t have this mathematical element in my Word) if you use that sign the equations are mathematically correct, but since r is not the same in eqs 9 or 10 you should write r1 in eq 9 and r2 in eq 10, and in the following discussion show that they vary - but are approximately the same.
If you do these changes, I may recommend the paper to be published
The following errors may then disappear:
line 260: maybe it should be (60 v)2.
line 263: t is not defined, maybe you mean r ?
Some other comments:
You can keep fig 2, but it should be made bigger to show the difference from a cirle
line 161: A reference to Eq 5 for E. Is necessary. Is this also from the Spencer model?
line 194: Equation 6 with a constant distance to the Sun is not correct and is not from the Spencer model. For each position on the Earth, you can get the distance to the Sun with the Horizon interface, available from JPL. Equation 6 have errors of several percent and may be replaced by a table. Example of precise calculations is found in this publication:
Rodolfo G. Cionco and Willie W.-H. Soon 2017. Short-term orbital forcing: A quasi-review and a reappraisal of realistic boundary conditions for climate modeling, Earth-Science Reviews., 166, 206-222.
Figure 8 shows the variation of the Earth’s obliquity during millenia and you see it is quite close to 23.5 for centuriea. The wiggle you mention is neglible for the solar insolation calculations.
Here you also find a discussion that using days as a unit for x (your equation 1) instead of the orbital longitude, you introduce errors of the order ±5% in the daily mean insolation. And your day count can be up to one day wrong.

Author Response
The Answer can be found as an attached file in word

Round 4
Reviewer 1 Report
Comments and Suggestions for Authors
I have reviewed the manuscript and find it improved for publication with some minor editing as I have outlined in the attached document.

No comments
Author Response
August 4, 2024
Referee comments to Article Rueda et al. Sun declination and distribution of natural beam
irradiance on Earth, version 3, received Aug. 1, 2024
Conclusion: The paper is now improved, and I can recommend publication with some minor
editing – see comments below.
Thank you. We will not leave any concern behind. Moreover, any comment which requires a correction or a further explanation… will be taken seriously into account, to improve our proposal.
In the cases where a comment cannot be executed, we will try to explain the reason in full.
You pointed us two typos , one sentence of undefined variables, one Figure whose footnote was not part of the Figure itself; but also at least a “three of a kind” shortage of explanations which might have led to a misunderstanding. We tried our best answering to those concerns, and added most explanations to the manuscript.
My comments
Section 2.3 Defining and suitable units
The authors use mathematical symbols, but use for angles arcdeg, arcmin and arcsec instead
of the mathematical symbols in the list above.
We deleted a word from our 2.3 heading, because it did not make sense
That is correct! We choose writing the units, as opposed to using symbols. That decision made clearer the titles of the visuals …
The unit abbreviations arcdeg, arcmin and arcsec are advised in cases where an analogous system appears in the same document. As we use minutes for time in the Equation of time; arcmin should not be mistaken when used together with minutes of time.
2.4 Declination cycle
Line 160: .. omega corresponded to a shift of declination to the north (winter or spring). Here you should write: spring and summer northern hemisphere
The confusion arises because seasons were not specified as boreal/austral (northern vs southern based). The signs of ω, as well as those of ∝ and δ, are available in Table 1. Our analysis is based in boreal seasons; while the actual version of the manuscript informs that the given seasons are boreal seasons.
The signs of ω pertaining to every particular season are mentioned in two occasions in the paragraph 2.4; we will improve writing in both mentions, to gain quality in the whole paragraph. The referred sign (the one you ask for) belongs exclusively to ω in those two mentions; so that we also separated all cases of positive parameters in a single paragraph to better explain the cases. The velocity of declination (ω) arise naturally as a negative or positive record from the derivative dδ/dt.
The negative records of ω arise naturally from the derivative dδ/dt for the summer and autumn [boreal] seasons …
The positive records of ω arise naturally from the derivative dδ/dt for the winter and spring [boreal] seasons …
Beyond this summarized description, a more detailed explanation is now provided in the paragraph itself
They probably mean that sign of declination (delta) shifts from negative to positive. This
happens at vernal equinox and last through spring and summer, while omega changes from
positive to negative at the summer solstices and will be negative summer and autumn.
The authors should mention that summer and winter is reversed on the southern hemisphere.
THIS last observation was the cause for the misunderstanding; some further explanation was required and it was given accordingly.
Line 217; although ? occurs to the left of ?, and ? occurs to the left of ?, by a distance equal to the length of an entire season, Fig 1 has time on the x-axes, and “to the left” means “earlier” or “before”. This makes more sense when you write about action and reaction…
Done!
A wider explanation was also provided in this paragraph, while the time scale was taken into account just as suggested by you.
Line 218-219: ...The functions of ?, “60?", and “3600?” (which are numerically identical to ?, ? and ?) yield pseudo-sinusoidal curves that resemble each other in shape, amplitude and frequency (Fig 1). Since this is the only place ? and ? are mentioned – and they are not shown in Fig 1, this definition of new variable is not necessary
You are right! Thanks a lot… Big mistake.
Now the sentence is re-written, to prevent the mention of undefined variables.
Equations 9 and 10: You find a mathematical sign for numerical value in the table on previous page. Use of mathematical symbols makes the text more precise.
We asked a mathematician for guidance regarding the possibility of using bars to indicate numerical values in Eqs. 9 and 10 (as opposed to the already included verbal clarification). Our consultant told us that the bars identify mostly a number as positive, so that the bars were only meant to remove the sign from an algebraic term.
He explained us that the bars are commonly referred to as … Absolute values
Line 265: Figure 3. The tagging of Equinoxes is missing.
Now Fig 3 has the adequate tags: solstices and equinoxes can be identified.
Please notice that the positions of the Tropic of Cancer and Capricorn become inverted in Fig 3.
Line 276: in Figs 2 and 3, the association between ?, ?, and ? and is clear: |?| ∝ 1/|?|, |?| ∝ 1/|?|, so that |?| ∝ |?|.
A word deleted because it made no sense. Bars for absolute values were added… as indicated in brown in the annotation above.
This relation means that ? approaches infinity when ? → 0 at the equinoxes – this is wrong
The given formal proportionality declarations are only meant to clarify whether a pair of parameters vary in direct or in inverse proportion to each other. The symbol provided only indicates relations of the type:
y∝1/x, when y can be expressed as Y = k/x
y∝x ​, when y can be obtained as Y = k x
The ranges on which this situation occurs is also specified for the case of our defined parameters. The associations only consider the actual records ranging between 0 and 29, for all three parameters; and now we also specified that these association are better understood when considering the associations as absolute values. The range of the parameter is also specified, and its relation to de circumferences is mentioned.
Line 473 and 473, monotonically accelerative throughout summer and winter ....
This probably means Northern hemisphere. But the acceleration is negative during summer, so if you mean positive acceleration it happens winter and spring northern hemisphere
WE analyzed the rationale behind the sentence 473 to 475. “The annual cycle solar meridian declination behaves monotonically accelerative throughout summer and winter, and monotonically decelerative throughout spring and autumn”. This is related to a concept named resultant drive, also denoted net drive. This concept is given at first mention in the 4th paragraph of the section 2.2.
This paragraph was written just now, to best explain our line of thought, thankyou…
The result of our analysis indicates that the sentence must remain just as it was written in the version of the manuscript that you revised. This one was the most challenging to explain. But a simplified rule will save us from getting lost in the way. Let’s define first an accelerative net drive (speeding up) as opposed to a decelerative net drive (slowing down), then we will go season by season to clarify when the net drive becomes accelerative or decelerative within every particular season.
Decelerative net drive: occurs when the product (ω)(α) yields a negative sign, otherwise stated, the signs of ω and α oppose.
Accelerative net drive: occurs when the product (ω)(α) yields a positive sign, otherwise stated, the signs of ω and α coincide.
|
Season |
δ |
ω |
α |
Sign of [(ω)(α)] |
α |
|
Spring |
+ |
+ |
- |
- |
Decelerative |
|
Summer |
+ |
- |
- |
+ |
Accelerative |
|
Autumn |
- |
- |
+ |
- |
Decelerative |
|
Winter |
- |
+ |
+ |
+ |
Accelerative |
As a general rule we noticed that the seasons synchronized with sections of the solar declination cycle (analemma) where the declination travels from the Equator to either Tropic, are Decelerative seasons. Conversely, accelerative seasons occur when the solar meridian declination moves from a Tropic to the Equator. The signs’ rules are a known result in Physics.
Please check the sentences and paragraphs which deal with this concepts…
Lines 176 – 185 , This lines were added just now in order to best explain ourselves
Lines 330 – 334
Lines 404 – 407
I must confess that this particular topic gave us a lot of headaches when we first took the concept in our writing. A straightforward approach to verify the reliability of our results consists in “ inverting all signs of all three parameters for the two hemispheres and notice that all associations remain true whether we took the southern hemisphere and its austral seasons as a guide “, instead of boreal seasons and the northern hemisphere . This rationale implies imagining all upside down.
Line 530: write Legacy instead of Le-Gacy Done!

Round 5
Reviewer 1 Report
Comments and Suggestions for Authors
The authors have responded very well to my suggestions and requirement.
I have only two minor remarks which can be fixed without another round of review:
line 239: the word gien has no meaning
Figure 2: the dotted line for Cancer and the dash-dot-dot-dash line for Capricorn are not visible
Finally I will congratulate the authors with persistence and willingness to answer all my remarks. Good Work.
Comments on the Quality of English Language
Well understandable for a non English reader.